# Nonwoven Textiles from Hyaluronan for Wound Healing Applications

**DOI:** 10.3390/biom12010016

**Published:** 2021-12-23

**Authors:** Jolana Kubíčková, Tomáš Medek, Jarmila Husby, Jana Matonohová, Hana Vágnerová, Lucie Marholdová, Vladimír Velebný, Josef Chmelař

**Affiliations:** Contipro a.s., Dolní Dobrouč 401, 56102 Donny Dobrucci, Czech Republic; Jolana.Kubickova@contipro.com (J.K.); Tomas.Medek@contipro.com (T.M.); Jarmila.Husby@contipro.com (J.H.); Jana.Matonohova@contipro.com (J.M.); Hana.Vagnerova@contipro.com (H.V.); Lucie.Marholdova@contipro.com (L.M.); Vladimir.Velebny@contipro.com (V.V.)

**Keywords:** hyaluronan, wet spinning, nonwoven textile, mechanical properties, wound dressing

## Abstract

Nonwoven textiles are used extensively in the field of medicine, including wound healing, but these textiles are mostly from conventional nondegradable materials, e.g., cotton or synthetic polymers such as polypropylene. Therefore, we aimed to develop nonwoven textiles from hyaluronan (HA), a biocompatible, biodegradable and nontoxic polysaccharide naturally present in the human body. For this purpose, we used a process based on wet spinning HA into a nonstationary coagulation bath combined with the wet-laid textile technology. The obtained HA nonwoven textiles are soft, flexible and paper like. Their mechanical properties, handling and hydration depend on the microscale fibre structure, which is tuneable by selected process parameters. Cell viability testing on two relevant cell lines (3T3, HaCaT) demonstrated that the textiles are not cytotoxic, while the monocyte activation test ruled out pyrogenicity. Biocompatibility, biodegradability and their high capacity for moisture absorption make HA nonwoven textiles a promising material for applications in the field of wound healing, both for topical and internal use. The beneficial effect of HA in the process of wound healing is well known and the form of a nonwoven textile should enable convenient handling and application to various types of wounds.

## 1. Introduction

Nonwoven textiles are porous layers formed from randomly oriented fibres of various diameters and lengths. The properties of these textiles are determined by those of the fibres themselves, by the layer topology (fibre entanglement and bonding) and other factors such as physicochemical interactions or the presence of adhesives. Nonwoven textiles are extensively used in various fields of medicine, among others, as wound dressings [1]. Recently, there has been a growing interest in the development of functionalized nonwoven wound dressings, e.g., with antibacterial properties [2,3,4] or with components that promote wound healing [5]. However, these wound dressings are still based on classical nondegradable textiles (e.g., cotton or synthetic polymers such as polypropylene).

Hyaluronan (HA) is an anionic, nonsulphated linear polysaccharide that is naturally present in the human body and is thus biocompatible, biodegradable and nontoxic. The monomer unit of HA consists of D-glucuronic acid and N-acetyl-D-glucosamine that are linked by alternating *β*(1→4) and *β*(1→3) glycosidic bonds. HA plays an important role in many biological processes including wound healing and tissue regeneration, hydration and lubrication. Due to its properties, HA is widely used in medicine, including wound-healing applications [6,7]. HA was used for the coating of nonwoven cotton fabrics [5,8] intended as wound dressings and carbon nonwoven fabrics for the repair of osteochondral defects [9]. The next step would be to prepare a nonwoven wound dressing with HA as the main component or even composed solely of HA.

Synthetic polymer fibres are commonly manufactured using melt-based processes. However, polysaccharides such as hyaluronan cannot be processed in the molten state. Consequently, polysaccharide fibres are usually prepared using solution-based processes. One of these is wet spinning, where the polymer is first dissolved in a suitable solvent and the obtained solution is then extruded through a nozzle into a coagulation bath (nonsolvent), where the fibres are formed [10].

The process of wet spinning can be used to prepare fibres suitable for processing into nonwoven textiles [11]. Fibres and nonwoven textiles from HA were prepared and modified with cell-adhesive peptides with the resulting composite fabrics proposed as scaffolds for tissue engineering [12] or cell-adhesive materials [13]. Nonwoven textiles prepared from cellulose derivatives, HA and etamsylate were tested in vivo as resorbable haemostatic materials, where the role of HA was to promote the subsequent healing of tissues [14]. If we look at HA derivatives, nonwoven textiles from the total benzoyl esters of HA were developed [15] and proposed as scaffolds for tissue engineering, but such materials are not suitable for wound-healing applications due to the high degree of HA modification (100%). Such HA derivatives are hydrophobic, insoluble and exhibit very slow degradation, the latter having been shown in vivo for polymeric films from the same benzoyl esters of HA [16].

In this study, we present nonwoven textiles composed solely from unmodified HA as a promising material for applications in the field of wound healing. HA is biocompatible and biodegradable, has a high capacity for fluid/moisture absorption and its beneficial effect on the process of wound healing is well known [17]. The processing of HA into a nonwoven textile should enable convenient handling and application to various types of wounds (superficial or deep, small or large, with or without bleeding, etc.). Once placed on the wound, the textile gradually hydrates and forms a protective gel layer on the wound. The textile hydration can be tuned by the fibre morphology and HA molecular weight. To prepare such textiles, we developed a process based on the wet spinning of HA into a nonstationary coagulation bath, yielding staple fibres that were further shortened and subsequently processed into nonwoven textiles by wet-laid technology. The obtained HA nonwoven textiles were soft, flexible and paper like. Their mechanical properties and resilience, as well as uptake of fluids leading to swelling and subsequent dissolution, strongly depended on the microscale fibre structure, which could in turn be influenced by the fibre preparation conditions. This is a great advantage, as the textile properties can thus be tuned specifically for various wound types. Last but not least, we addressed the safety of the prepared textiles. Cell viability tests were carried out with two relevant cell lines (3T3, HaCaT), showing that the textiles are not cytotoxic, while the monocyte activation test (MAT) was carried out to rule out pyrogenicity.

## 2. Materials and Methods

### 2.1. Material

Sodium hyaluronate (HA), with weight-average molecular weights (*M*_W_) of 0.8, 1.2 and 2.0 MDa, was provided by Contipro a.s. Pure 2-propanol was obtained from Brenntag. Calcium chloride dihydrate was obtained from Penta s.r.o. Sodium dihydrogen phosphate and disodium hydrogen phosphate were obtained from Lach-Ner. Agar (for microbiology) was obtained from Fluka. Tert-butanol, sodium azide, HPLC grade 2-propanol, gelatine from bovine skin (gel strength ~225 g Bloom, type B) and all materials for cell cultivation experiments were obtained from Sigma-Aldrich.

### 2.2. Nonwoven Textile Preparation

The preparation of nonwoven textiles was based on the wet-spinning technology according to a patented procedure [18]. The used workflow is schematically illustrated in Figure 1. The polymer solution was prepared by dissolving HA in demineralized water to a concentration of 10 mg/mL. The polymer was dissolved at room temperature for 16–24 h using a magnetic stirrer, obtaining a homogenous, transparent and well flowing viscous solution suitable for spinning. This solution was extruded through a spinning nozzle (blunt steel needle; inner diameter—0.65 mm; length—50 mmL) into a coagulation bath. The design of the spinning apparatus is schematically shown in Figure 2. The speed of dosing was regulated by a NEXUS 6000 syringe pump to 0.6 mL/min per nozzle. The coagulation bath contained absolute 2-propanol and was nonstationary, i.e., with a circulating flow, where the flow rate was regulated and could be varied. The composition of the coagulation bath slightly changed over time: there was a small yet gradual increase in water content due to water being introduced into the bath with the HA solution and not being removed. The actual 2-propanol content was determined by measuring the density of the coagulation bath. If the 2-propanol content decreased to less than 90%, the spinning process was stopped and the bath regenerated.

The fibres of suitable diameters were obtained by precipitation in the coagulation bath. To produce enough fibres for one nonwoven textile, 30 mL of the HA solution had to be processed. Subsequently, the fibres were separated using separation combs (see Figure 2) and placed into a maturation bath to increase their mechanical properties. The fibres were first subjected to cutting using a Microtron MB 800 mixer (Kinematica AG, Malters, Switzerland) at 12,000 rpm for 30 s (total volume of the dispersion was 400 mL) and then left to maturate in absolute 2-propanol for 16 to 20 h. In the maturation bath, the system of hydrogen bonds between hyaluronic acid chains that has been broken by dissolving the polymer is reconstituted. This leads to the restoration of the supramolecular structure of hyaluronic acid and to the improvement of the mechanical properties of the fibres [18]. The next step was filtration. The fibre suspension was poured into an 8 × 8 cm steel frame with a porous substrate (knitted polyamide fabric on a steel mesh) and excess 2-propanol was filtered using a vacuum to achieve short filtration times. The obtained layer of wet fibres on the polyamide fabric substrate was then transferred to a slightly curved heated plate (50 °C), pressed to the plate with a cotton woven textile to prevent deformations of the textile during drying and left to dry for up to 30 min. After drying, the resulting nonwoven textile was put into a Steripack packaging and stored at standard laboratory conditions.

### 2.3. Basic Characterization of HA Nonwoven Textiles

The first analysis of a prepared nonwoven textile was weighing on an analytical balance (Mettler Toledo, Columbus, OH, USA). From the obtained mass of the sample in mg, the mass per area in mg/cm^2^ was calculated, as this property is independent on the size of the textile and thus enables an easier comparison both between individual samples and with materials reported in other studies.

Dry matter content was measured using a Q500 thermogravimetric analyser (TA Instruments, New Castle, DE, USA). The textile sample (10 mg) was placed in a platinum pan, heated to 105 °C (10 °C/min) and kept under isothermal conditions for 15 min. Synthetic air (flow rate 60 mL/min) was used as the measuring atmosphere. The dry matter content was calculated from the measured weight loss. Each sample was measured in duplicate and the results averaged.

The content of residual 2-propanol in the textiles was determined using gas chromatography with a headspace autosampler and a mass spectrometer detector (Agilent Technologies, Santa Clara, CA, USA). The textile sample (10 mg) was inserted into a vial and 2 mL of water and 0.1 mL of internal standard (tert-butanol, 0.1 mg ml^−1^ in water) were added. After equilibration (shaking, 90 °C for 60 min), the sample was injected into a DB624-UI column (Agilent J&W GC Columns) and the concentration of 2-propanol was determined. Each sample was measured in duplicate and the results averaged.

The porosity of the textiles was measured using a 3Gz Porometer (Quantachrome Instruments, Boynton Beach, FL, USA) and Porofil wetting fluid. The measured area of the textile sample was 3.14 cm^2^ (diameter 2 cm) and the thickness of each specimen was measured using a mechanical micrometre. The parameters used for data evaluation were: shape factor—1, size factor—0.64 and pore tortuosity—1.

### 2.4. Molecular Mass Determination

Molecular weight data were measured using size exclusion chromatography with multi-angle laser light scattering detection (SEC-MALLS). The analyses were performed on a Waters Alliance liquid chromatograph (Model e2695, Waters Corporation, Milford, MA, USA) equipped with a differential refractometer Optilab rEX and DAWN TREOS multi-angle laser light scattering photometer (both from Wyatt Technology Corporation, Santa Barbara, CA, USA).

The sample was prepared by weighing a certain amount of nonwoven textile and its dissolution in the mobile phase. The final concentration depended on the expected *M*_w_ of HA analysed, i.e., the sample concentration ranged from 0.2 to 0.5 mg ml^−1^ for HA with *M*_w_ ranging from 2.0 to 0.8 MDa. Each sample was filtered through an Acrodisc Syringe Filter (0.45 µm, 25 mm diameter) with the Supor membrane (Pall). The injection volume was 100 µL. The separation was carried out using PL aquagel-OH 60 (7.5 × 300 mm, 15 µm) and PL aquagel-OH mixed-H (7.5 × 300 mm, 8 µm) columns connected in series and thermostated at 40 °C. The mobile phase was 0.1 M sodium phosphate buffer (pH adjusted to 7.5) + 0.05% sodium azide at a flow rate 0.8 mL min^−1^. All chemicals for SEC-MALLS were HPLC grade and the mobile phase was filtered through a Nylaflo Nylon Membrane Filter (0.2 µm, 47 mm diameter; Pall).

Data acquisition and molecular weight calculations were performed using the ASTRA 8 software (Wyatt Technology Corporation, Santa Barbara, CA, USA). The specific refractive index increment d*n*/d*c* of 0.155 mL g^−1^ was used for HA [19].

### 2.5. Microscopy

The morphology of the prepared nonwoven textiles was studied with an Olympus LEXT OLS5000 3D Measuring Laser Microscope (Olympus, Tokyo, Japan). The fibre diameter measurements were performed with the Olympus Stream Desktop software.

### 2.6. Measurement of Mechanical Properties

Tensile properties of the nonwoven textiles were measured using an Instron 3343 single-column testing system with a 100 N head (Instron, Norwood, MA, USA). Textile samples (8 × 1 cm) were measured at 23 ± 2 °C and at a relative humidity of 50 ± 5%. The samples were pre-tensioned at 0.05 N and then stretched at a rate of 30 mm min^−1^ until break. The strength at break (MPa) and strain at break (relative elongation, %) were evaluated.

The thickness of samples used for mechanical analysis was measured using a VL-50 mechanical thickness meter (Mitutoyo, Kawasaki, Japan). Each textile sample (8 × 1 cm) was measured at 5 points and the results averaged.

### 2.7. Testing of Application Properties on a Simulated Wound Surface

Since the nonwoven textiles are intended for clinical application such as wound dressing, it was desirable to study their application properties such as handling, hydration and dissolution in a relevant in vitro experiment, i.e., on a simulated wound surface. The used model was based on the work of Andrews and Kamyab [20], who developed an in vitro model for the measurement of surgical dressing adherence.

Wound surfaces were simulated using a gel consisting of gelatine, agarose and CaCl_2_. First, 10 mL of 0.2M CaCl_2_ was prepared by dissolving 0.294 g of calcium chloride dihydrate in 10 mL of demineralized water. Next, 16 g of gelatine and 4 g of agar were placed into a beaker, followed by the addition of 396 mL of phosphate buffer and 4 mL of 0.2M CaCl_2_. The obtained solution was subjected to microwave heating (550 W) with occasional stirring until it started boiling for the second time (here, boiling means that foam started to develop, like when heating milk). Subsequently, the still-hot solution was dosed in 30 mL volumes into Petri dishes (diameter 9 cm), using an injection, and left to solidify overnight at ambient temperature. This procedure enabled us to prepare 400 mL of the gel (13 gel-filled Petri dishes).

A real wound is covered by proteinaceous wound exudate, the proteins of which might interact with the applied material. Therefore, to better model the properties of wounds, the gel surface was moistened with human plasma (500 µL per Petri dish, distributed equally on the gel by a plastic laboratory spatula). Plasma was obtained by centrifugation (10 min, relative centrifugal force 1300× *g*) of whole noncoagulated venous blood collected from a volunteer into BD Vacutainer lithium heparin tubes.

Two types of tests were performed. In the first one, textile samples (4 × 4 cm) were applied on the moistened simulated wound surface and the time from the first contact of the sample with the simulated wound to its full hydration was measured. In the second test, the possibility of repositioning the sample after application was tested (handling test). Immediately after the sample (4 × 4 cm) came into full contact with the simulated wound surface, the sample was peeled off completely and applied back without delay, simulating the repositioning of the product by a surgeon after unsatisfactory placement.

### 2.8. In Vitro Cytotoxicity (MTT Assay)

To evaluate the cytotoxicity of HA nonwoven textiles, the MTT assay was performed with 3T3 (mouse fibroblasts) and HaCaT (human keratinocytes) cell lines. The testing was carried out in line with ISO 10993-5 [21]. Dulbecco’s Modified Eagle Medium supplemented with 10% fetal bovine serum was used as the cultivation medium. The textile samples were dissolved in the cultivation medium at a concentration of 1000 µg/mL. This solution was applied both in the original concentration and after dilution. The cells were seeded into 96-well plates at a concentration of 3000 and 5000 cells per well for 3T3 and HaCaT cells, respectively. After overnight incubation, the cells were treated with the sample solution in 1000, 500, 100 and 10 µg/mL concentrations. Pure cultivation medium was used as the negative control. Viability was measured by the MTT (3-(4,5-dimethylthiazol- 2-yl)-2,5-diphenyl-tetrazolium bromide) assay after 24, 48 and 72 h of incubation. MTT stock solution (5 mg/mL; 20 µL) was added to the cell culture medium in each well and the plates were incubated for 2.5 h at 37 °C. Subsequently, the MTT solution was removed, 220 µL of a lysis solution (2-propanol, 10% dimethylsulfoxide, 10% Triton X-100) was added and lysis was performed for 30 min at room temperature and 150 rpm. Finally, the absorbance at 570 and 690 nm (*A*_570_ and *A*_690_, respectively) was read by an EnSight Multimode Reader (PerkinElmer, Waltham, MA, USA). The final absorbance *A* was calculated as:(1)$$A=A_{570}-A_{690}$$

The change in the viability of the treated cells was calculated relative to the negative control at 0 hrs (*A*_CTRL,0_) according to the following equation:(2)$$\mathrm{Viability}\;\mathrm{relative}\;\mathrm{to}\;\mathrm{control}\;\left[\%\right]=\frac{A}{A_{\mathrm{CTRL},0}}\times 100$$

If the relative cell viability is above 70% even for the highest concentration, the tested sample can be considered as noncytotoxic [21].

### 2.9. Monocyte Activation Test

The effect of the HA nonwoven textiles on immune cell activation was assessed using the Monocyte Activation Test (MAT). The textile samples were dissolved in normal saline (0.9% NaCl) at a concentration of 1 mg/mL. Heparinised whole blood from four healthy donors was pooled and 10× diluted with normal saline. The diluted blood was placed in sterile microtubes and 100 µL of the sample solution was added, followed by incubation at 37 °C for 16 to 20 h. In parallel, the inner control sample was prepared by further adding Reference Standard Endotoxin (RSE, Merck, Darmstadt, Germany, cat. no. #E0150000) at a concentration of 0.25 EU/mL to the sample solution. The purpose of the inner control sample was to exclude undesired interaction between the sample and the RSE lipopolysaccharide. For evaluation purposes, also solutions of pure RSE at concentrations of 10, 5, 1, 0.25 and 0.1 EU/mL were tested.

After incubation, the microtubes were manually shaken and centrifuged at 13 000 rpm for 10 min. The upper phase (plasma diluted in the saline) was collected for IL-6 analysis using an IL-6 Human Uncoated ELISA kit (Invitrogen, Waltham, MA, USA, #88-7066-88) according to the manufacturer’s protocol. Absorbance was read by an EnSight Multimode Reader and the Kaleido software (both from PerkinElmer, Waltham, MA, USA) was used for evaluation.

## 3. Results and Discussion

In this section, the basic characteristics of HA nonwoven textiles will first be described, followed by the analysis of the effects of two key parameters on the textile structure and properties. These two parameters are the *M*_W_ of HA and the flow rate of the nonstationary coagulation bath. Finally, the cytotoxicity and pyrogenicity of the textiles will be assessed in vitro.

To evaluate the flow regime in the circulating coagulation bath, we calculated the Reynolds number for the individual flow rates. The Reynolds number (*Re*) is the ratio of inertial forces to viscous forces and was calculated as:(3)$$Re=\frac{\rho \;v\;d}{\eta }$$
where *ρ* is the fluid density, *v* is the fluid velocity, *d* is the inner diameter of the tube (8 mm) and *η* is the dynamic viscosity of the fluid. In a tube, the flow is laminar for *Re* < 2300, transitional for 2300 < *Re* < 4000 and turbulent for *Re* > 4000. The fluid velocity was calculated from the used flow rate, the fluid density was measured using a DMA 35 density meter (Anton Paar, Graz, Austria) and the dynamic viscosities were taken from the literature [22]. The used flow rates and the calculated *Re* are presented in Table 1.

### 3.1. Basic Characterization

Nonwoven textiles from HA have the appearance of white self-supporting porous sheets (Figure 3, left) that are soft, flexible and paper like. They enable convenient manipulation and can be repeatedly folded without breaking (Figure 3, right). Similar to paper, they can be easily torn or cut to the desired shape.

The mass per area of the textile depends on the amount of polymer used for its preparation. Using the procedure and dosing described in Section 2.2, the obtained HA textiles had a mass per area of 4.8 ± 0.45 mg/cm^2^.

The dry mass content in the textiles was measured using thermogravimetry. The average content of dry mass was 83.3 wt.% with all measured values being between 82.7 and 84.4 wt.%, showing good reproducibility. The dry mass content depends on the final drying conditions and thus did not depend on the flow rate (*Re*), as was expected.

The residual content of 2-propanol, the only organic solvent used in the textile preparation, was measured in three textiles prepared under various conditions (*Re* of 1950, 3150 and 4250). The residual content of 2-propanol was always below 0.01 wt.%, which was the limit of quantification (LOQ) of the used method. It is important that the prepared textiles do not contain higher amounts or residual organics since they could lead to toxicity [23].

### 3.2. Effect of Flow Regime during Coagulation

Nonwoven textiles were prepared from 1.2 MDa HA using various flow rates of the coagulation bath inside the coagulation tubes (see Table 1). The influence of the resulting flow regime on the textile morphology is illustrated in Figure 4. Laminar and transitional flows result in the coagulation of the spinning solution mostly into polydisperse fibres with diameters between 0.5 and 10 μm. In both cases, the resulting nonwoven textiles have porous structure, which is reflected by short filtration times needed to remove excess 2-propanol during the preparation of these textiles (Table 2) and their strength at break is relatively low (see Figure 5) as their breaking is mostly the result of mutual movement of fibres opposed only by frictional forces. The turbulent flow of the coagulation bath results in the coagulation of the spinning solution mostly into the form of amorphous precipitate with a limited amount of actual fibres. The resulting nonwoven textiles have significantly lower porosity reflected by the long filtration times (Table 2) and their strength at break is higher with greater variability (see Figure 5), because their breaking is the result of actual fracture along a sharp edge initiating from the random point of weakness of the nonuniform precipitate layer as opposed to mutual movement of intertwined elements.

The dependence of the textile strength and strain at break on *Re* are depicted in Figure 5. As discussed above, the strength at break generally increased with increasing *Re* due to the presence of amorphous precipitate that reinforces the textile. The variability of the results, demonstrated by the standard deviation, also increased with increasing *Re*. The strain at break first increased with increasing *Re* between 1600 and 3500 and then slightly decreased at the two highest *Re* values. The increase is due to the reinforcing effect of the amorphous precipitate domains that bind adjacent fibres and thus increase the overall cohesion. However, as the amount of amorphous precipitate become higher, its presence hinders the rearrangement of the material during stretching and thus leads to lower strain at break values.

### 3.3. Effect of HA Molecular Weight

The effect of HA molecular weight was studied using samples at 0.8, 1.2 and 2.0 MDa. Nonwoven textiles were prepared from these samples at various Re, namely 1950, 3150 and 4250. The dependence of the textile structure and properties on *Re*, which was described in detail for 1.2 MDa HA in the previous section, was also observed for the other two samples. A laminar and transient flow led to textiles with fibrous structures, which was reflected by short filtration times during their preparation, while turbulent flow led to larger amounts of amorphous precipitate in the textiles and longer filtration times.

Changes in HA *M*_W_ were not reflected in the macroscopic mechanical properties represented by the strength and strain at break, partly due to the larger variability of the results in general. However, increased *M*_W_ resulted in larger fibre diameters, as shown in Figure 6 (distribution of fibre diameters) for Re 1950. The distribution seems narrower for the lowest *M*_W_ of 0.8 MDa, while that measured for 2.0 MDa is broader, with a plateau-like region between 1 and 4 µm. The distribution for the middle *M*_W_ of 1.2 MDa is between these two cases. Note that analogous trends were observed for the other *Re* values, although evaluation becomes more complicated as the content of amorphous precipitate increases (Figure 7). Analogously, an increase in fibre diameter with *M*_W_ was reported for electrospun polyvinylalcohol [24] and silk fibroin [25]. This trend is mostly explained by the pronounced effect of *M*_W_ on the rheological properties of the spinning solution [26] and also on the chain entanglement [27].

Larger fibre diameters should lead to higher porosities of the fibre layer during filtration and thus result in shorter filtration times [28]. In our experiments, significantly shorter filtration times were observed for the *M*_W_ of 2.0 MDa irrespective of *Re*, but the differences in fibre size between 0.8 MDa and 1.2 MDa were not reflected in filtration process (Table 3).

### 3.4. Testing of Application Properties on a Simulated Wound Surface

To provide some basic assessment of HA nonwoven textile application properties in the context of their intended use (see below), such as handling and hydration, we performed two experiments on a simulated wound surface (human plasma moistened gelatine-agarose hydrogel). In the first experiment, the time to full hydration after placing the textile on the simulated wound surface was measured. The second test studied the possibility of repositioning the sample after application (handling test).

Let us first comment on the intended use of the textiles. HA is extremely hydrophilic and readily dissolves or swells (forms a gel) in contact with water, depending on the amount of fluid in contact with the polymer. Once placed on the moist wound surface, the textile will gradually hydrate and form a protective gel layer (wet wound dressing). Subsequently, the biological effects of HA can come into play and promote the healing process [17]. The form of a nonwoven textile enables easy manipulation and ensures a homogeneous distribution of HA over the entire wound area.

The results of both application tests are summarized in Table 4. It can be observed that the time to full hydration increased significantly with increasing HA *M*_W._ This outcome is in agreement with our experience with the solubility of HA and similar literature [29]. This increase was observed for textiles prepared using laminar (*Re* 1950), transient (*Re* 3150) and turbulent (*Re* 4250) flows during fibre coagulation. If the individual flow regimes are compared, it is evident that the textile hydration is slower at lower *Re*, with larger differences observed between turbulent and transient flows than between transient and laminar flows. We assume that this effect is related to textile porosities. It was reported that liquid transport is faster in fabrics with smaller pores [30], provided thorough the wetting of the material by the liquid. This suggests that the amorphous precipitate containing HA textiles obtained at higher *Re* should be hydrated faster because they have smaller pores, as can be seen in the microscopic images in Figure 4 and Figure 7.

To verify the assumed differences in porosity, we carried out measurements of the mean pore size for the same samples that were used for the application property tests. The results are summarized in Table 5. The mean pore size decreases with increasing *Re* and with decreasing *M*_W_, in agreement with the discussion above, with the effect of *M*_W_ being small when the flow is turbulent (*Re* 4250).

Our results show that the integrity of the textile during its repositioning increased with an increasing *M*_W_ of HA and increasing *Re*. The effect of *M*_W_ is related to the slower hydration (dissolution) of HA with higher *M*_W_, while the effect of *Re* is related to the increased strength (mechanical resilience) of textiles prepared using turbulent flow in the coagulation step (see Figure 5). Apart from better handling, the use of a high *M*_W_ of HA is also beneficial with respect to its biological properties that contribute to wound healing, e.g., high *M*_W_ HA is anti-inflammatory, anti-angiogenic [31] and analgesic [32]. In summary, it seems beneficial to use a higher *M*_W_ of HA for preparing nonwoven textiles for wound-healing applications.

### 3.5. In Vitro Evaluation of Cytotoxicity and Pyrogenicity

The cytotoxicity of HA nonwoven textiles was assessed using the MTT assay on two cell lines, 3T3 and HaCaT. The results for various extract concentrations and incubation times are shown in Figure 9 and Figure 10 for 3T3 and HaCaT cells, respectively. Viabilities were evaluated relative to the controls at 0 h, i.e., values above 100% correspond to increased cell proliferation, while values below 100% correspond to inhibited proliferation. One can see in Figure 9 and Figure 10 that the cells were either activated or showed proliferation similar to controls, with all values safely above the threshold limit of 70%. Nonwoven textiles from HA can thus be considered noncytotoxic [21].

The ability of HA nonwoven textiles to elicit inflammatory response (and its pyrogenicity) was assessed using MAT [33]. For the HA textile sample, IL-6 was below the quantification limit (<2 pg/mL) (Figure 11) showing that it did not induce an inflammatory response. The pyrogenicity was evaluated as lower than the pyrogenicity of 0.1 EU/mL but should be re-evaluated when the exact dosing and the length of the potential treatment is established.

## 4. Conclusions

We prepared nonwoven textiles from HA using a process combining wet spinning and wet-laid textile technology. The obtained HA nonwoven textiles were soft, flexible and paper like. Their mechanical properties, handling and hydration depended on the microscale fibre structure, which could in turn be influenced by the flow regime in the coagulation bath. Laminar and transitional flows lead to softer textiles composed from polydisperse fibres, while turbulent flow leads to textiles containing high amounts of amorphous precipitate, resulting in lower porosity and higher strength. This is a great advantage, as the textile properties can be tuned specifically for various applications. Importantly, the noncytotoxicity and nonpyrogenicity of the textiles were verified in vitro. These properties of HA nonwoven textiles make them promising for use in medicine, especially in wound-healing applications. The beneficial effect of a high *M*_W_ HA on wound healing is well known and the form of a nonwoven textile should enable convenient handling and application to various wound types.

## Figures and Tables

**Figure 1 biomolecules-12-00016-f001:**
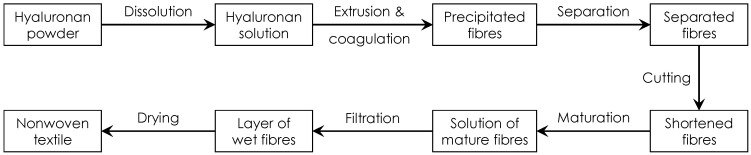
Scheme depicting the sequence of main production steps during the preparation of a nonwoven textile from HA.

**Figure 2 biomolecules-12-00016-f002:**
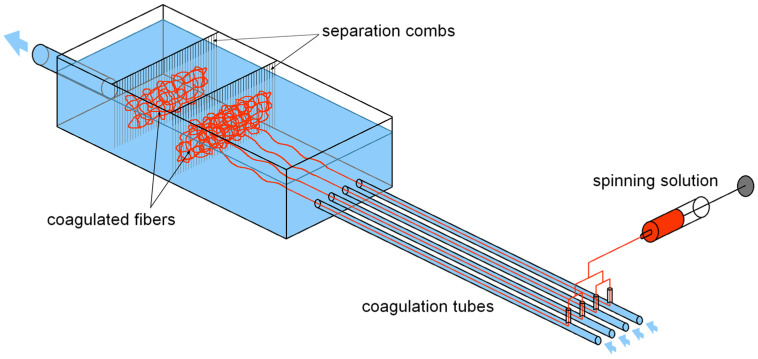
Schematic depiction of the spinning apparatus. The HA solution is dosed into 4 nozzles, each placed in a separate coagulation tube, perpendicular to the coagulation bath flow. The tubes lead into a larger vessel with separation combs, which are used to separate the coagulated fibres from the circulating bath.

**Figure 3 biomolecules-12-00016-f003:**
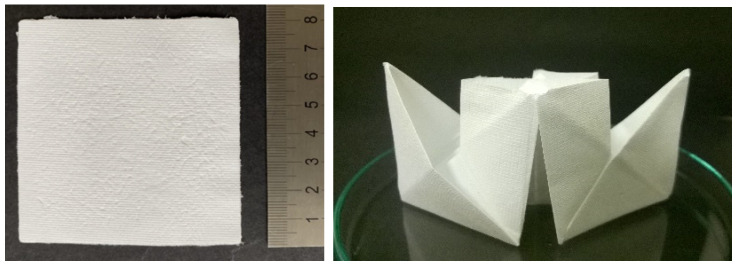
Photograph of an 8 × 8 cm nonwoven textile (**left**) and of the same textile folded into a paper ship (**right**), illustrating that the textile can be easily folded without breakage.

**Figure 4 biomolecules-12-00016-f004:**
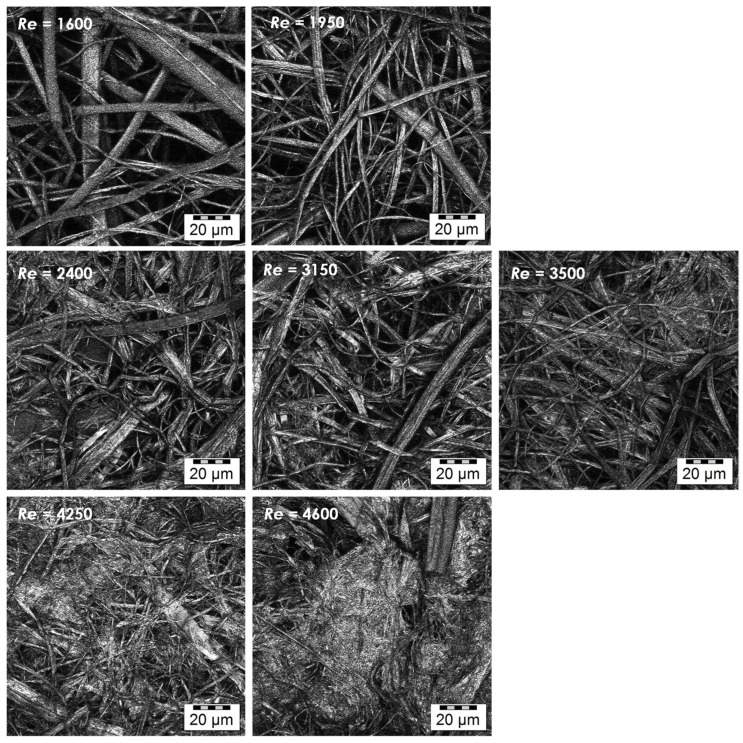
Microscopic images of the textile microstructure for all studied *Re*. (Olympus LEXT OLS5000 3D Measuring Laser Microscope, MPlanApo N 100 x/0.95 LEXT; Olympus, Tokyo, Japan).

**Figure 5 biomolecules-12-00016-f005:**
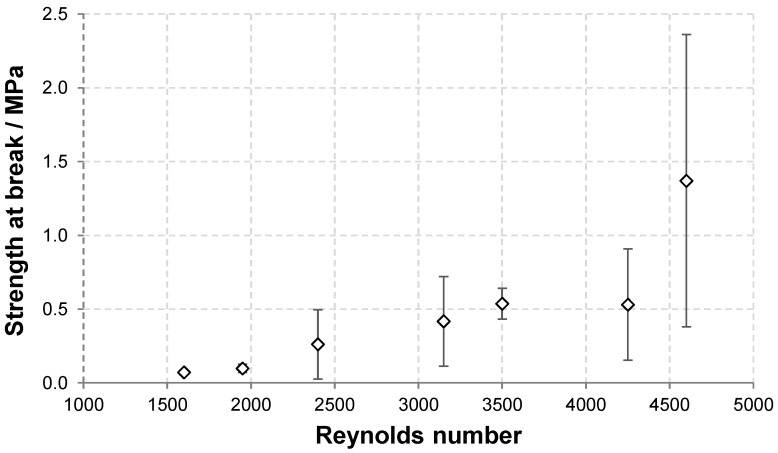
Dependence of the strength and strain at break of HA nonwoven textiles on *Re*.

**Figure 6 biomolecules-12-00016-f006:**
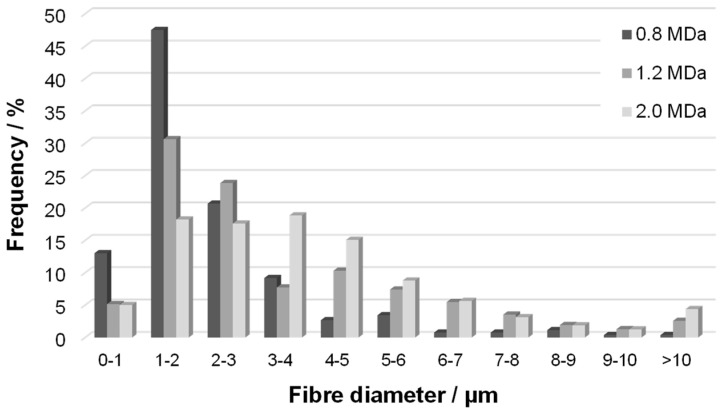
Distribution of fibre diameters (frequency) in nonwoven textiles prepared at *Re* 1950 from HA samples with various Mw.

**Figure 7 biomolecules-12-00016-f007:**
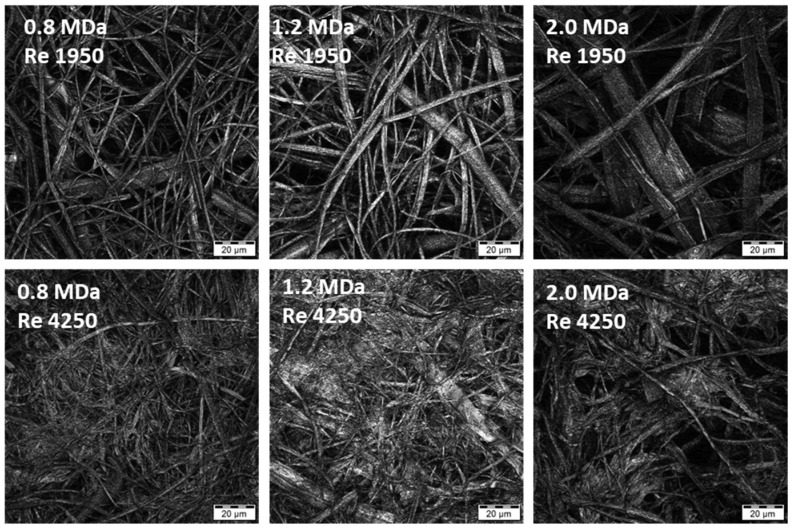
Microscopic images of the textile microstructure for HA *M*_W_ of 0.8, 1.2 and 2.0 MDa and two selected *Re* (laminar and turbulent flow). (Olympus LEXT OLS5000 3D Measuring Laser Microscope, MPlanApo N 100 x/0.95 LEXT).

**Figure 8 biomolecules-12-00016-f008:**
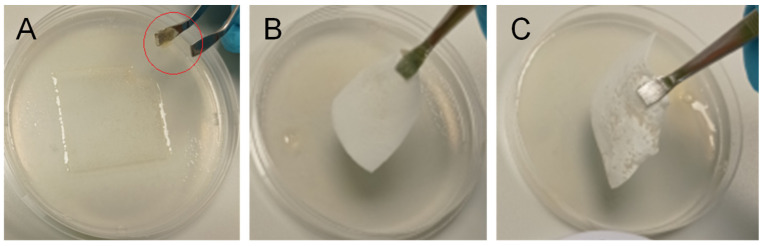
Repositioning of HA nonwoven textiles after their placement on a simulated wound surface (hydrogel): (**A**) textile could not be removed from the surface due to the loss of mechanical properties (*M*_W_ 0.8 MDa, *Re* 1950); fragment of the textile on the tweezers is marked by a red circle; (**B**) textile could be removed from the surface, but its edges bended (*M*_W_ 1.2 MDa, *Re* 1950); (**C**) textile could be removed from the surface without bending, making its repositioning possible (*M*_W_ 2.0 MDa, *Re* 4250).

**Figure 9 biomolecules-12-00016-f009:**
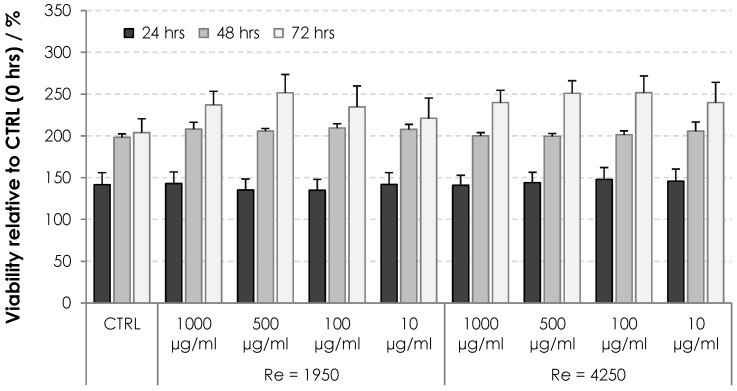
Viability of 3T3 cells subjected to dissolved HA textiles for 24, 48 and 72 h. Columns represent mean values and error bars the standard deviation.

**Figure 10 biomolecules-12-00016-f010:**
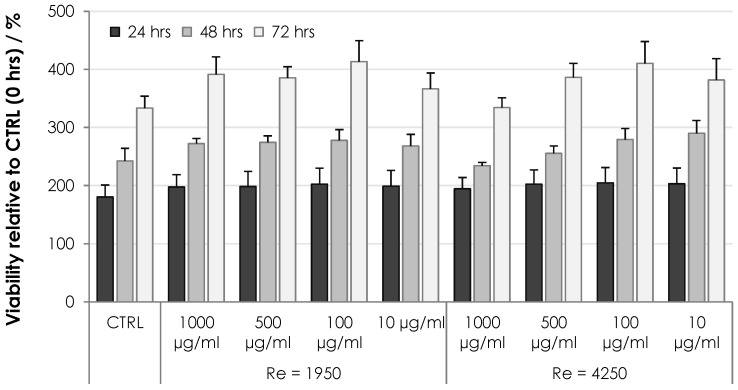
Viability of HaCaT cells subjected to dissolved HA textiles for 24, 48 and 72 h. Columns represent mean values and error bars the standard deviation.

**Figure 11 biomolecules-12-00016-f011:**
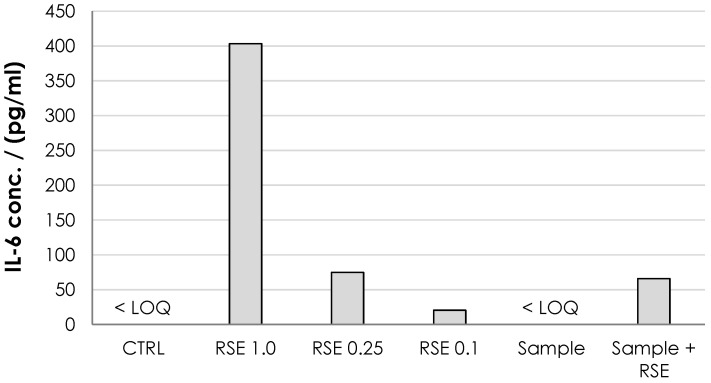
Monocyte activation expressed as the IL-6 concentration. The negative control (normal saline, CTRL) and tested dressing solution (sample) were below detection limit for IL-6 ELISA (<LOQ; 2 pg/mL). Three solutions of endotoxin standards (RSE) were used as positive controls at 1.0, 0.25 and 0.1 EU/mL. The standard addition of RSE at 0.25 EU/mL to the sample was used to assess the potential inhibition of MAT by the sample.

**Table 1 biomolecules-12-00016-t001:** Overview of the studied flow rates, fluid velocities and the relevant *Re* values.

Flow Rate/(L/h)	*v*/(m/s)	*Re*
430	0.619	1600
530	0.760	1950
630	0.909	2400
830	1.195	3150
930	1.337	3500
1130	1.625	4250
1230	1.766	4550

**Table 2 biomolecules-12-00016-t002:** Average filtration times obtained for the individual *Re* values.

*Re*	Flow	Filtration Time/min
1600	laminar	0:22 ± 0:03
1950	laminar	0:31 ± 0:01
2400	transitional	0:31 ± 0:01
3150	transitional	0:47 ± 0:02
3500	transitional	1:06 ± 0:03
4250	turbulent	7:40 ± 1:41
4600	turbulent	30–120

**Table 3 biomolecules-12-00016-t003:** Representative filtration times (step of fibre suspension filtration during the textile preparation) measured for various *Re* and *M*_W_ of HA.

*Re*	Filtration Time/min 0.8 MDa	Filtration Time/min 1.2 MDa	Filtration Time/min 2.0 MDa
1950	0:23 ± 0:02	0:31 ± 0:01	0:06 ± 0:00
3150	1:31 ± 0:07	0:47 ± 0:02	0:23 ± 0:01
4250	6:17 ± 0:15	7:40 ± 1:41	1:17 ± 0:07

**Table 4 biomolecules-12-00016-t004:** Results of application tests on a simulated wound surface.

Sample	Time to Full Hydration	Possibility of Repositioning
0.8 MDa + laminar flow (*Re* 1950)	20 s	Repositioning not possible: immediate tearing upon manipulation (Figure 8A)
1.2 MDa + laminar flow (*Re* 1950)	180 s	Repositioning not possible: removed in one piece, but edges bend (Figure 8B)
2.0 MDa + laminar flow (*Re* 1950)	360 s	Could not be completely removed, tearing during removal (approx. half)
0.8 MDa + transient flow (*Re* 3150)	16 s	Repositioning not possible, tearing during removal (approx. half)
1.2 MDa + transient flow (*Re* 3150)	90 s	Repositioning not possible, tearing during removal (approx. half)
2.0 MDa + transient flow (*Re* 3150)	300 s	Repositioning not possible, tearing during removal (approx. half)
0.8 MDa + turbulent flow (*Re* 4250)	10 s	Repositioning not possible: immediate tearing upon manipulation
1.2 MDa + turbulent flow (*Re* 4250)	80 s	Repositioning not possible: removed in one piece, but edges bend
2.0 MDa + turbulent flow (*Re* 4250)	100 s	Repositioning possible: removed in one piece, without bending (Figure 8C)

**Table 5 biomolecules-12-00016-t005:** Results of mean pore size measurements.

	Mean Pore Size/µm		
*M*_W_/MDa	*Re* 1950	*Re* 3150	*Re* 4250
0.8 MDa	6.10	3.12	1.57
1.2 MDa	9.07	6.18	1.54
2.0 MDa	18.59	13.07	1.78

## Data Availability

Not applicable.

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
