# Peer review of "Nonwoven Textiles from Hyaluronan for Wound Healing Applications"

_biomolecules, 2021, doi:10.3390/biom12010016_

Round 1

Reviewer 1 Report

The present research paper shows originality in a well-presented manner. All the parts are well presented and deeply discussed.

Author Response

Response in the attached document.

Reviewer 2 Report

The article titled “Nonwoven textiles from hyaluronan for wound healing applications” may be a useful contribution to the journal; however, few changes should be taken into consideration

  1. In Figures 9 and 10, What is the fate of Control (untreated cells) in both 3T3 cells and HaCaT cells? The author mentioned in the Y-axis “viability relative to control/%”, I recommend adding the control values in each figure and represent Y-axis as “Cell viability (%)”. Please modify the figures (Figure 9 & 10) accordingly. It is very difficult to come into the conclusion without the control results.
  2. According to the objective of this research work “Nonwoven textiles from hyaluronan for wound healing applications”. The pH stability of the finally prepared material is a big concern when it is prepared for wound healing applications. I am very much concerned about the synthesized material and its stability behavior in the wound environment, particularly for a period of 3-7 days. Please include “Stability studies” is as part of the characterization of the synthesized materials. So I strongly recommend performing the above-mentioned studies in different pH conditions (which is more appropriate to the wound environment) in PBS buffer in order to bring out a clear conclusion behind the material stability and discuss the outcomes of this manuscript.
  3. There is no proper discussion was made in section 3. I strongly recommend adding suitable references from the literature to support their claims in each paragraph under section 3. Eg. Section 3.1; Section 3.2; and Section 3.5.

Author Response

Response in the attached document.

Reviewer 3 Report

Dear Authors

The presented work in your manuscript is interesting for the readers. However, a major revision is needed before reconsidering your manuscript for publication.

The specific comments are listed below.

Comments to the authors

Section 2.7:

A) The authors have to mention the reference of this test.

B) The characterizations of the simulated wound surface such

as roughness, Zeta potential, and the AFM analysis are missing. These characterizations are essential to compare the simulated wound surface with the real one. The authors have to provide and use in the discussion of the obtained results.

C) The authors mentioned that " Plasma was obtained by centrifugation (10 min at 1300 g) ?. I think the authors mean (10 minutes at 1300 rpm). Please correct. 

Section 2.8: Mentioned references. 

Section 2.9: Mentioned references. 

Section 3. Results and discussion

3.3: The authors selected the last Re of both laminar and traditional flow, while they selected the first Re in the Turbulent flow. Please explain.

A) In figure 6, it is noticed that the frequency (%) indicate maximum values between 1-2 um of fibres diameters for 0.8 MDa and 1.2 MDa, while a plateau was recognized between 1-4 um of fibres prepared using 2.0 MDa. The authors need to give an explanation.

B) In Table 3, the authors emphasise that " The difference in fibre size between 0.8 MDa and 1.2 MDa were not reflected in filtration". This is correct for the results of Re 1950 and 4250. However, at Re 3150, the difference is quite clear where the filtration time reduced from 1:31 to 0:47 minutes. The fibres diameters distribution of the fibres prepared at Re 3150 may explain the obtained result. The authors have to introduce the results of fibres diameter prepared at both Re 3150 and 4250 for comparison and discussion.

Section 3.4:

A) The authors again selected only two groups of fibres prepared under laminar and turbulent flow and omitted the fibres prepared under transitional flow. WHY?!!! 

B) The authors explained the shortening of the time of full hydration as a result of using 2.0 MDa by the porosity resulting from the formation of HA precipitated particles. The authors ignored the surface area which may be the determining factor in the hydration process. Data of the surface area, pore size and pore size distribution in addition to the total pore volume are essential in explanation of the obtained behaviour. Authors should add to the manuscript. 

Section 3.5: References are required to support the presented results in Figures 10 and 11.

C) 

Author Response

Response in the attached document.

Round 2

Reviewer 3 Report

Dear Authors

Thanks for your responce to the comments satisfactory.

I can recommend your revised manuscript for publication.

Author Response

Thanks for your review.